# Survey of Missouri Landowners to Explore the Potential of Woody Perennials to Integrate Conservation and Production

Raelin Kronenberg [1,2,*], Sarah Lovell [1], Bhuwan Thapa [3], Christine Spinka [4], Corinne Valdivia [5], Michael Gold [1] and Sougata Bardhan [2]

[1] School of Natural Resources, University of Missouri Center for Agroforestry, Columbia, MO 65201, USA; sartaylov@gmail.com (S.L.); goldm@missouri.edu (M.G.)

[2] Department of Cooperative Research, College of Agriculture, Environmental and Human Sciences, Lincoln University of Missouri, Jefferson City, MO 65101, USA; bardhans@lincolnu.edu

[3] Department of Geography and Planning, College of Arts and Sciences, Appalachian State University, Boone, NC 28608, USA; thapab@appstate.edu

[4] Department of Health Management & Informatics, University of Missouri School of Medicine, Columbia, MO 65212, USA; spinkac@missouri.edu

[5] College of Agriculture, Food and Natural Resources, University of Missouri, Columbia, MO 65201, USA; valdiviac@missouri.edu

* Correspondence: rlk5hp@mail.missouri.edu

**Abstract:** The state of Missouri, USA offers a unique opportunity for tree planting under several federal conservation programs. However, many landowners remain hesitant to enroll and take land out of agricultural production. This study explores the willingness of landowners to adopt agroforestry systems with food producing tree and shrub species through federal conservation program funding using mail and online surveys. Surveys followed the Dillman Tailored Design Method to collect data on landowners' farm characteristics, production practices, and land management choices. Survey participants were sampled on a county basis within each of the six major geographic regions of the state. Twelve counties were randomly selected, and surveys were mailed to a proportional sampling of farm addresses gathered from each of the county tax assessor offices. The goal of the survey was to (1) identify landowners' current land management practices and goals, (2) understand landowners' perceptions of and preferences for different planting plans for their farm, and (3) capture landowners' interest in participating in conservation programs to assist in the planting of trees and shrubs on their land. Our analysis of this survey found that landowners are receptive to agroforestry plantings, rating them higher on average than traditional agricultural land management practices. Landowner age, past participation in a conservation program, and presence of marginal land all had significant correlation with willingness to adopt agroforestry. The inclusion of technical assistance or federal conservation funding was found to increase the willingness of landowners to plant multifunctional agroforestry designs.

**Keywords:** conservation; multifunctional; land use; agroforestry; survey; photo elicitation

## 1. Introduction

To better understand the importance of incorporating new approaches to agricultural land management for conservation, it is helpful to reflect on how agriculture and conservation goals have been reconciled throughout history. This paper focuses on the United States, a country where agricultural production has been and continues to be an important part of the national economy. The intensification of agricultural production in the United States to achieve higher production has led to a decrease in soil productivity through nutrient depletion, topsoil erosion, and water pollution [1–4]. These negative effects are primarily due to the use of repeated plowing and soil disturbance during row cropping. Soil degradation was severely exacerbated during the year 1933, when the U.S. experienced the Dust

Bowl. The removal of trees and grass field margins to maximize agricultural production contributed to the widespread drought and severe topsoil erosion across the western portion of the country. In response to the unprecedented loss of soil, the U.S. government initiated the Great Plains Forestry Project, which supported the planting of 220 million trees across the contiguous U.S. [5]. Despite this program's success in reintroducing trees into the agricultural landscape and reducing wind erosion across the plains, the benefits were not permanent. Many of the tree rows were eventually removed as agricultural producers once again sought increases in production. Farm resource concerns continued to grow and were worsened during the U.S. farm financial crisis of the 1980s. Larger investments in agriculture grew the U.S agricultural export market and encouraged farmers to shift to larger-scale, more intense production methods [6]. The upscale in agricultural production encouraged greater removal of many of the windbreaks and other soil protection strategies implemented during the Great Plains Forestry Project. With the expansion in agricultural production efficiency came a saturation of the market and decrease in the value of many of the commodities being produced [6,7]. The U.S. Government responded to the development of these economic and resource concerns by establishing a growing number of conservation policies and economic support programs under the 1985 to 2002 farm bills [8] (Table 1).

**Table 1.** Timeline of important events leading to the establishment of federal conservation programs, 1933–2002.

| Year | Event | Outcome |
| --- | --- | --- |
| 1933 | Dust Bowl | Severe soil erosion, abandoned farms |
| 1934 | Great Plains Shelterbelt Project | Planting of 220 million trees in windbreaks across the U.S. to address soil erosion from the Dust Bowl era |
| 1935 | Soil Conservation Service Established | Program to provide funding to farmers for soil conservation practices |
| 1975 | Secretary of Agriculture encourages farmers to plant 'fencerow to fencerow' | Reversal of previous conservation gains, leading to unsustainable farming practices |
| 1980s | U.S Farm Financial Crisis | Falling farm income leads to shift to intense production practices on larger areas of land |
| 1985 | CRP Funded | Program to retire ecological sensitive/marginal land from agricultural production |
| 1994 | Soil Conservation Service Renamed Natural Resources Conservation Service | Shift in government support for conservation beyond soil and crop productivity |
| 1996 | EQIP Funded | Program to promote agricultural production, forest management, and environmental quality as simultaneously compatible goals |
| 2002 | CSP Funded | Program to encourage conservation practices that support whole-farm resource goals |

Source: Adapted from [9].

While the federal conservation programs listed in Table 1 had some success in addressing both economic and environmental concerns throughout the decades [10,11] shifts in political support, agricultural markets, and land management preferences continue to leave vulnerable acres of land in intensive production [12,13]. Farmers need for productive uses of their land [14] along with the inflexible management requirements, low program

payments, and a complex administrative process leave many landowners with little interest in enrolling in conservation programs [9,15]. These programs also experience issues with backlogs of unfunded applications and budget pressures that make it difficult to accept all landowners who are interested in enrolling [16]. Additionally, the amount of land enrolled in CRP has been steadily decreasing from a peak of 36.8 million acres in 2007 [17] to 20.6 million acres in 2021 [18]. Farmland that was once enrolled in a conservation program is being returned to production once their federal contracts expire instead of remaining in perennial cover [13,17]. The reasons landowners decide to resume planting commodity row crops such as corn and soy include shifts in commodity prices and restoration of land production capacity through improved soil health post-CRP enrollment [13].

*Agroforestry Adoption*

An alternative to conservation programs that encourage the management of land separately for conservation or production is to establish multifunctional approaches for land use. The benefits of land management that integrates conservation goals while producing marketable goods or services include improved soil health, wildlife habitat, and income generation for landowners [1,19]. Agroforestry, or the intentional planting and management of trees with crops and/or livestock [3,20,21], is a management strategy that can support both the conservation and production goals of farms. Despite these benefits, we do not see broad adoption of agroforestry in agricultural landscapes. Researchers have studied the process of landowner decision-making and agroforestry adoption factors. There are several common variables shown to influence agroforestry adoption in the United States. These include landowner capacity, (i.e., landowners having the knowledge and ability to adopt agroforestry), landowner attitudes towards conservation and trees, awareness of the defined agroforestry practices, and landowner farm characteristics such as acreage and income [22–24]. Even with the increased awareness of agroforestry adoption factors, barriers still exist.

To build an understanding of how to direct long-term conservation initiatives for multifunctional plantings of trees and shrubs, we considered the values and opinions of landowners in Missouri, a state in which agroforestry practices have been promoted and supported. We used a state-wide survey to gather information about farm goals, practices, and interest in planting trees through conservation program funding, and landowners' interest in agroforestry. We also capture landowners' interest in conservation programs more broadly, as well as their willingness to plant agroforestry using conservation program resources. This research will provide further insights into landowners' acceptance of productive conservation by exploring the following research questions:

1. What are landowners' perceptions of and preferences for different planting plans that include agroforestry for their farm?
2. To what extent do landowners show interest in participating in conservation programs to assist in planting trees and shrubs on their land?
3. How do landowner characteristics and land use goals influence their decision to plant agroforestry on their farm?

## 2. Materials and Methods

### 2.1. Study Site

The importance of agricultural activities, as well as the diversity of different enterprises (e.g., row crops, livestock, and specialty crops), for the state of Missouri presents an interesting case study on how conservation and production can be integrated throughout agricultural landscapes. The state's geography ranges from prairie land in the north to the Ozark highlands in the south. Part of the Mississippi river floodplain area in the southernmost region of Missouri, termed the bootheel, Is the state's most intensively row-cropped area. In total, 95,000 farms across Missouri cover 27.8 million acres of land, roughly two thirds of the state [25]. The agricultural industry contributes $88 billion to the state's economy and employs over 400,000 people [25].

A strategic approach to contacting Missouri landowners to participate in the survey was used to recruit a representative sample. For this process, Missouri was divided into six geographic regions used by the University of Missouri Extension offices to explore any differences in landowners' agroforestry preferences due to their farms' geographic location and local climate (Figure 1). All counties were designated either rural or urban based on the classification schemes determined by the county rurality data from the 2010 Census Data [26]. Twelve representative counties, one rural and one urban from each region, were randomly chosen. Sample data was selected using minimum sample size estimation and sample size with finite population formulas [27]. We used proportional sampling of the total number of farms in each of the regions to determine the number of addresses to randomly select from each county. A minimum sample size of 50 was set to ensure enough survey responses for statistical analysis. Due to difficulty in getting addresses from all the counties' tax accessor offices that were selected to be included in the survey, two of the geographic regions, southeast and west central, relied on samples from a single county. Following a similar protocol as Barbieri and Valdivia [28] and Mattia et al. [29], lists of all agricultural land parcels with the landowner contact information were procured from each county's tax assessor office. These contact lists were edited to remove absentee landowners, businesses, and county land. A proportional sample from each address list was then randomly selected to attain a 95% confidence level and the chosen contacts were mailed the survey.

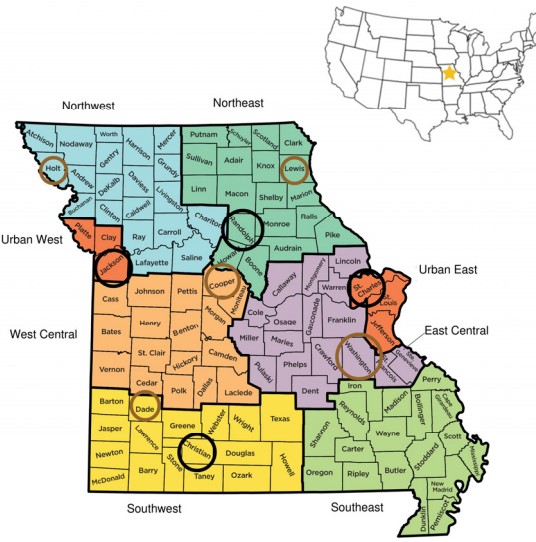

**Figure 1.** The six regions of Missouri as divided by MU Extension were used to compare planting design preferences. Counties included in the sample are indicated by dots. Brown dots mark the rural counties; black dots represent urban counties.

### 2.2. Survey Instrument and Timeline

The survey questionnaire began with a series of screening questions. These included asking if the participant was the primary decision maker for their land and if they are over the age of 18 (see Supplementary Materials). The main body of the survey instrument contained four sections (Table 2). The first section collected information about the farm, including its location by county, acreage, presence of marginal land, and the landowner's goals. The second section included detailed planting plans and perspective-view digital renderings of a field, pasture, riparian zone, and forest scene, which were used to capture landowners' preferences for the different landscapes [30,31]. Each of the planting plans varied in complexity from a landscape under typical management (open row crop field, open cattle pasture, grass filter strip, and forest), to a simple agroforestry or timber production design (conifer windbreak in a field, hardwood silvopasture with cattle, basic riparian buffer, timber stand), or to the multifunctional agroforestry plantings (multifunc-

tional windbreak in a field, pecan silvopasture with cattle, multifunctional riparian buffer, and a forest farm) (Figure 2). All planting designs adhere to NRCS conservation practice standards for species selection and spacing. Likert scale ratings for each design allowed participants to indicate their preference for the different planting plans. The survey also asked landowners about their agreement/disagreement with several statements including the profitability of the planting, the challenge of maintenance, and the conservation benefits for each of the multifunctional agroforestry designs. After the design ratings, questions on conservation program participation and land use were included. Participants could freely explain their choice to enroll or not enroll in a conservation program. The final section collected basic demographic information to allow for cross-referencing respondents with census data to check representativeness of the sample for Missouri landowners.

**Table 2.** Overview of survey sections sent to landowners in Missouri.

| Section | Question Summary |
|---|---|
| (1.) Farm information | Farm location, acres owned, acres leased, farm experience, presence of marginal land, farm goals |
| (2.) Planting Plan Ratings | Rate the desirability of each plan, indicate agree/disagree to comments on planting plans |
| (3.) Land use | Land use/management activities, conservation program enrollment, interest in conservation program—free response |
| (4.) Demographic Information | Age, occupation, education, income |

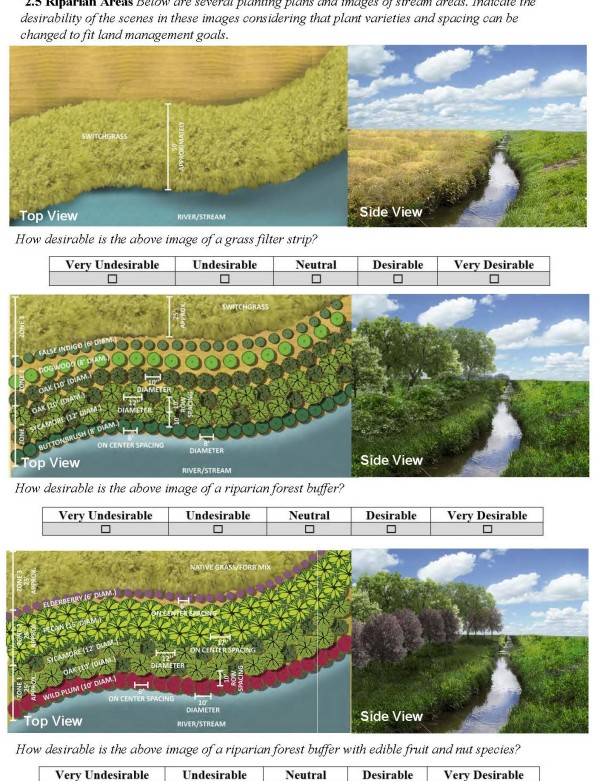

**Figure 2.** Sample of planting plan number 2.5 from the survey instrument showing a grass filter strip, a basic riparian forest buffer, and a multifunctional riparian forest buffer. This same layout was used to ask participants about the desirability of silvopasture, windbreak, and a forest farm. Photo elicitations prepared by Isaac Palomo. Survey prepared by Raelin Kronenberg. Please see Supplementary Materials for complete survey instrument used.

The survey questionnaires were sent out using a modified Dillman Tailored Design Method [32] via mail on 30 April 2021 and 21 May 2021, and addresses were processed for mailing. A link to the online survey using the Qualtrics platform was included on the paper copies. The participants were offered the opportunity to be entered into a draw to win a $25 gift card by completing the survey before a specified deadline. They could choose to remain anonymous or share their contact information for the gift card drawing. A total of 3673 surveys were sent out between April and May. Due to an initially low response rate of less than 3%, 3035 additional surveys were sent out on 26 July 2021 to help increase the total number of responses (Figure 3). Reminder postcards were mailed on 26 July to the first two survey rounds. Reminders for the additional round of addresses were mailed on 16 August.

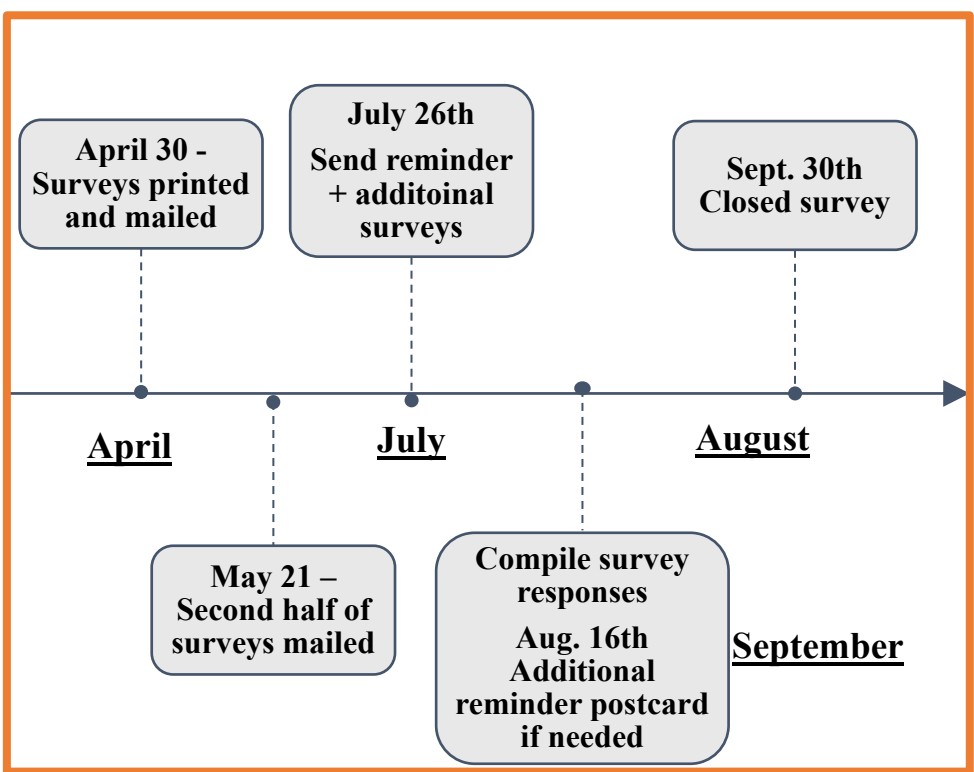

**Figure 3.** Project timeline for sending surveys and collecting responses.

*2.3. Statistical Analysis*

Descriptive statistics were performed in SAS software version 9.4 (SAS Institute, Cary, NC, USA). The desirability ratings for each planting design were predicted using a two-way analysis of variance (ANOVA) based on each of the designs and the survey respondent. A one-way ANOVA in SAS proc General Linear Model (GLM) was used to predict the landowners' willingness to plant each agroforestry design based on a single predictor. We began analysis by using each demographic variable as an independent factor (age, gender, farming as primary occupation) and again in separate models using each of the farm goals (income, conservation, recreation, education, agritourism, and lifestyle) as the independent factor. Future interest to participate in a conservation program was also measured and used to predict willingness to plant agroforestry. Post-hoc testing was performed using Fisher's Least Significant Difference to examine significant differences between agroforestry design ratings. Free-response questions and comments about the planting designs and conservation programs were sorted and explored separately using NVivo software version 12, released 2017 (QSR International, Burlington, MA, USA). Summaries of these responses are included in the discussion to enrich the survey's quantitative findings.

## 3. Results and Discussion

### 3.1. Survey Response

Of the combined 6708 surveys sent out, 366 responses were collected. After accounting for undeliverable addresses, we had a response rate of about 6%, which is lower than expected for survey research [33]. This is likely due to the over-surveying of landowner populations, reducing their interest in participating [34]. The average age of survey respondents was 61 years old; the majority self-identified as male (71.6%) and white (96.5%). Approximately 75% had some education above a high school diploma, with 45% earning a college degree. The majority are not full-time farmers (73%), and when asked about their primary occupation, "retired" was the most common answer with 60 responses. Working in healthcare (7), education (9), finance (8), and in local county government positions (7) were other listed primary occupations. The most common farm net income was none ($0) to less than $20,000 a year. The 2017 National Agricultural Statistics Service (NASS) can be used to compare how representative the sample of Missouri's farming population is [35] (Table 3).

**Table 3.** Missouri Farmer population statistics compared to survey sample.

| Population | Average Age | Gender Male | White | Primary Farmer | Average Farm Size | Average Income & Income Range |
|---|---|---|---|---|---|---|
| State | 57.4 | 63.8% | 98.4% | 39% | 291 | $12,649 |
| Survey Sample | 61.2 | 71.6% | 96.5% | 27% | 197 | <$20,000 |

### 3.2. Planting Plan Preferences

By comparing the mean ratings of the three planting plan levels in each scene, we can see which of the designs is rated higher among respondents. The multifunctional agroforestry designs had the highest mean rating scores across each scene, so we can infer that they were preferred over the plans that represented typical agricultural land management practices (Figure 4). These findings are similar to other survey and interview research findings, indicating that landowners generally support multifunctional agroforestry planting designs [29,36]. This study's findings mirror a wider shift in landowner preferences towards multifunctional land management that includes agroforestry, which is especially promising compared to earlier adoption studies that found farmers had little to no interest in agroforestry [28,37]. Despite a growing preference for multifunctional planting designs, when asked about their willingness to plant each agroforestry system, landowners indicated they are unsure of whether they would plant the presented designs on their farm. In other adoption studies, farmers also indicated that they were hesitant to establish agroforestry practices including alley cropping and riparian forest buffers with fruit and nut trees [23,28,36,38].

While the landowners indicated there are benefits from agroforestry practices, especially for supporting conservation including wildlife habitat, protecting natural resources, and reducing soil erosion, they also expressed concerns over the costs to establish and maintain these plantings. Comments on the multifunctional agroforestry designs highlighted concerns over the lack of knowledge on managing the agroforestry plantings. The absence of management skills and the technical knowledge required to successfully adopt agroforestry has been a reoccurring theme throughout agroforestry adoption literature [29,36,39,40]. Many landowners also indicated that agroforestry practices would not be profitable. They expressed concerns about the cost of establishing and maintaining agroforestry plantings and the absence of developed markets for fruit, nuts, and other specialty products produced by the species in these plantings. Other researchers have found a recognized need for more developed markets and infrastructure to support agroforestry adoption [23,29].

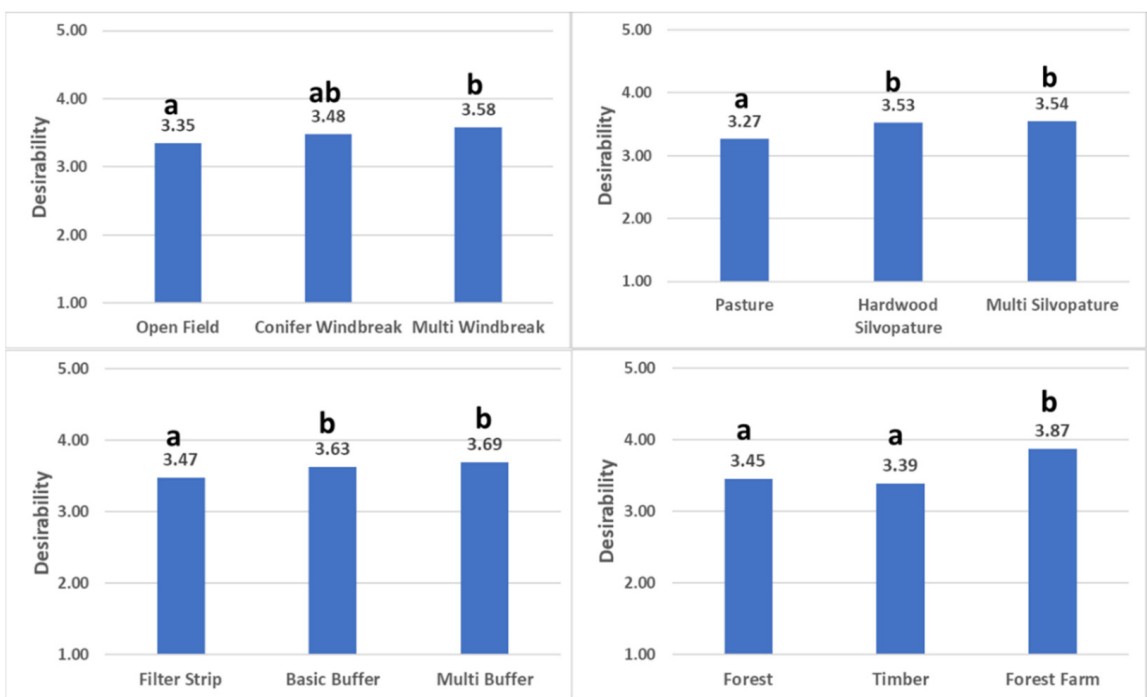

**Figure 4.** Comparison of mean desirability ratings between planting plan images in section two of the survey using a two-way analysis of variance (ANOVA) based upon each of the designs and the survey respondent. Different letters indicate means that are statistically significantly different. 1 = Very Undesirable, 3 = Neutral, and 5 = Very Desirable on Likert Scale shown to participants.

In addition to general concerns over financial returns and management requirements, landowners commented on the large area of land that several of the designs would require. Participants explained that the riparian forest buffer and multifunctional windbreak require a lot of space in and along fields, meaning only larger farms would have the land to plant these designs. Available acres are an important factor for agroforestry adoption because landowners with more land available to transition to alternative management are more willing to plant agroforestry [22–24,41]. Other comments from respondents indicated that the designs did not apply to their land; several landowners did not have a stream, field, forest, or pasture on their property and were thus unable to consider planting the design in question.

After examining the desirability ratings of the planting designs across all responses, we explored if there were any differences in the ratings between urban and rural counties and across the six geographic regions. For nearly all planting designs, there was no statistically significant difference between their average ratings of desirability in urban versus rural counties, suggesting preferences for planting designs are not related to population density. Only one planting plan, the conifer windbreak, had a slightly higher desirability rating by landowners in urban counties compared to rural ones (see Table 4). This could be due to the benefits provided by the windbreaks to urban areas such as visual screening, wind protection, and odor control [42,43]. Comparing the different regions of Missouri, planting plans also had similar desirability ratings (Table 5). The forest farm design was the only plan with significantly different ratings between the various regions. Forest farms were rated higher in the state's east central (EC) and northeast (NE) regions than others. We did not gather information in the survey to determine the reason for this difference, but it could be because of the major metropolitan area, St. Louis, within this region. The proximity of farms in the EC region to St. Louis likely increases their connections to diverse urban markets, therefore providing opportunities to grow and sell specialty forest products [23,24]. The higher ratings of the forest farm in this area are something to explore with additional

surveys or landowner interviews to better understand the potential of forest farming for this area.

**Table 4.** Mean Desirability Ratings of Planting Designs between Urban and Rural Counties.

| Design | Urban | Rural | *p*-Value |
|---|---|---|---|
| Field | 3.19 | 3.42 | 0.1006 |
| Conifer Windbreak | 3.68 | 3.41 | 0.0433 * |
| Multi Windbreak | 3.77 | 3.51 | 0.1146 |
| Pasture | 3.08 | 3.35 | 0.1006 |
| Hardwood Silvopasture | 3.62 | 3.49 | 0.3508 |
| Multi Silvopasture | 3.65 | 3.49 | 0.2773 |
| Filter Strip | 3.46 | 3.5 | 0.7744 |
| Basic Riparian Buffer | 3.65 | 3.64 | 0.9491 |
| Multi Buffer | 3.71 | 3.69 | 0.9129 |
| Forest | 3.49 | 3.43 | 0.7039 |
| Timber | 3.33 | 3.41 | 0.5879 |
| Forest Farm | 3.90 | 3.86 | 0.749 |

Note: Table shows comparison of mean desirability ratings of planting designs between urban and rural counties using a two-way analysis of variance (ANOVA) based on each of the designs and the survey respondent. 1 = very undesirable, 3 = neutral, 5 = very desirable on a Likert Scale. * Indicates a significant difference between the mean desirability rating between counties.

**Table 5.** Mean Desirability Ratings of Planting Designs Between Regions of Missouri.

| Design | NW | NE | EC | WC | SW | SE | *p*-Value |
|---|---|---|---|---|---|---|---|
| Field | 3.49 | 3.19 | 3.33 | 3.66 | 3.20 | 3.86 | 0.2072 |
| Conifer Windbreak | 3.36 | 3.64 | 3.63 | 3.36 | 3.33 | 3.29 | 0.3541 |
| Multi Windbreak | 3.50 | 3.56 | 3.85 | 3.55 | 3.46 | 3.43 | 0.6647 |
| Pasture | 3.35 | 2.97 | 3.37 | 3.52 | 3.43 | 3.50 | 0.1083 |
| Hardwood Silvopasture | 3.57 | 3.68 | 3.59 | 3.52 | 3.22 | 3.43 | 0.2541 |
| Multi Silvopasture | 3.54 | 3.45 | 3.70 | 3.67 | 3.38 | 3.71 | 0.6212 |
| Filter Strip | 3.47 | 3.54 | 3.20 | 3.86 | 3.38 | 3.57 | 0.0787 |
| Basic Riparian Buffer | 3.53 | 3.74 | 3.51 | 3.86 | 3.57 | 3.29 | 0.4376 |
| Multi Buffer | 3.50 | 3.89 | 3.77 | 3.76 | 3.49 | 3.29 | 0.218 |
| Forest | 3.39 | 3.44 | 3.57 | 3.44 | 3.44 | 3.14 | 0.907 |
| Timber | 3.33 | 3.58 | 3.30 | 3.53 | 3.10 | 3.43 | 0.1818 |
| Forest Farm | 3.46 | 3.99 | 4.21 | 3.88 | 3.81 | 3.29 | 0.006 * |

Note: Table shows comparison of mean desirability of planting designs between the MU extension regions of Missouri using a two-way analysis of variance (ANOVA) based on each of the designs and the survey respondent. NW = Northwest, NE = Northeast, EC = East Central, WC = West Central, SW = Southwest, SE = Southeast. Ratings of 1 = very undesirable, 3 = neutral, 5 = very desirable on a Likert Scale. * Indicates a significant difference between the mean desirability ratings.

### 3.3. Conservation Program Interest and Participation

To improve our understanding of landowners' interest in conservation programs, we asked why they did or did not participate in them. Most of the respondents were not currently enrolled in any conservation program. When explaining why they chose not to participate, the primary reason was a lack of knowledge about the conservation programs available in their county. Broadly, landowners indicated they have little awareness and knowledge of the programs available to them, how to enroll, what the management activities entail, and ultimately knowing if they can provide the management required to establish the conservation practices and maintain enrollment. Earlier studies on landowner participation in conservation programs arrived at similar conclusions [29,44]. Other landowners had some sense of what conservation programs entailed, but they preferred their current management practices and saw no need to integrate new approaches into their production systems. Some landowners mentioned not owning enough acres to qualify for enrollment or indicated that their current land management practices support conservation. Several landowners noted they prefer to make their own decisions managing their land based on

their current knowledge (Table 6). Others explicitly stated they did not want government involvement on their land or farm. This distrust of the government has been highlighted in other research as a significant reason why landowners chose not to participate in conservation programs [9,45,46]. Lastly, old age and health concerns kept some participants from enrolling their land in conservation programs since they worried that they would be unable to do the work needed to implement and maintain the conservation practices. Other researchers have also found that age can influence willingness to invest in long-term conservation [12,29,47]. Although landowners chose not to participate in conservation programs, the majority (69%) indicated they are interested in enrolling in the future. We found landowners interested in enrolling in a conservation program in the future have a strong conservation ethic (Table 7). Other studies drew similar conclusions on the importance of landowners' conservation and stewardship values in their decision to enroll in conservation programs [14,29].

**Table 6.** Frequency of comments by landowners on why they do not want to participate in conservation programs.

| Comment Theme | Number of Comments |
| --- | --- |
| Lack of knowledge of programs | 66 |
| Programs not applicable to farm | 26 |
| Content with current management | 19 |
| Doesn't like conservation program requirements | 12 |
| Independent decision maker | 12 |
| Costs too much to participate | 10 |
| Did not qualify for the program | 10 |
| Participation takes too much time | 8 |
| Age and health prevent participation | 5 |
| Unprofitable to participate | 3 |
| Never thought about conservation programs | 2 |
| No help to establish conservation practices | 2 |
| Program contracts last too long | 1 |

Note: Table quantifies the number of comments on why landowners did not or would not participate in conservation programs. These comments are not verbatim to those made by participants, but are used as categories to group and quantify similar themes in the landowners' responses.

**Table 7.** Frequency of comments by landowners on why they want to participate in conservation programs.

| Comment Theme | Number of Comments |
| --- | --- |
| Want to participate to support conservation | 13 |
| Want to create wildlife habitat | 6 |
| Want to participate to address resource concern | 5 |
| Want to participate for financial benefits | 2 |
| Want more knowledge about conservation practices | 1 |

Note: Table quantifies the number of comments on why landowners chose to or would participate in a conservation program These comments are not verbatim to those made by participants, but are used as categories to group and quantify similar themes in the landowners' responses.

### 3.4. Factors for Agroforestry Adoption

Focusing on the factors for landowners' willingness to plant multifunctional agroforestry plantings can help guide future outreach initiatives and direct the work of conservation and natural resource professionals in the field. Age was found to be a significant factor in predicting willingness to plant the agroforestry designs, with older landowners (67+) being less willing than younger (under 35) and middle-aged landowners (36 to 66). This contradicts findings from other researchers, who found that age had no effect on interest in planting riparian buffers [36]. In contrast, Pattanayak et al. [41] found age to be a factor in adopting agroforestry, but it is not always significant. A reason older landowners may be hesitant to plant trees is due to the long period before the benefits of perennial

conservation practices are seen. Often, older landowners are more hesitant to commit time and money for plantings they are unlikely to be able to harvest and enjoy during their lifetime [22,29]. The aging farmer population presents a challenge to the widespread adoption of long-term conservation and agroforestry practices.

Other significant factors for predicting greater willingness to adopt agroforestry are the presence of marginal land on the farm and the landowner's interest in participating in a conservation program. Landowners with marginal land were more willing to plant the agroforestry designs than those who did not have marginal lands. This is consistent with previous research that found that the presence of marginal land was a motivator for landowners to enroll in a conservation program [29]. For the survey, marginal land was defined as less productive land than the average farmland in the participant's area. Marginal land presents resource concerns and is a management challenge due to erosion, poor soil productivity, and/or flooding [9,29]. We found many landowners have some amount of marginal land, including uneven, rocky ground prone to flooding or areas that are shaded. Addressing these concerns while producing additional benefits provides an excellent starting point for expanding conservation efforts while maintaining production [43]. We also found that respondents who indicated a future interest in participating in conservation programs are more willing to plant multifunctional agroforestry designs compared to those who are not interested in conservation programming. This aligns with other research findings [38,48]. We found that the number of acres the landowner owned, being a beginning farmer (having farmed for less than 10 years), one's primary occupation as a farmer, and gender were not significant factors in predicting willingness to plant agroforestry (Table 8).

**Table 8.** Demographic factors influencing landowner's willingness to plant agroforestry, where designs refer to each of the multifunctional agroforestry planting plans shown to participants of the survey.

| Design | Independent Factor [1] | *p*-Value | Model Fit | Variable Type [2] | Relationship [3] |
|---|---|---|---|---|---|
| Multi Windbreak | Age | 0.0029 ** | $r^2 = 0.0498$ | Categorical | Negative |
| | Gender | 0.065 | | Categorical | |
| | Primary Farmer | 0.3058 | | Categorical | |
| | Conservation Program Interest | <0.0001 ** | $r^2 = 0.0904$ | Categorical | Positive |
| | Farm Income | 0.0799 | | Categorical | |
| | Marginal Land | 0.1011 | | Categorical | |
| | Acres | 0.3123 | | Continuous | |
| | Beginning Farmer | 0.7897 | | Categorical | |
| Multi Riparian Buffer | Age | 0.0009 ** | $r^2 = 0.0590$ | Categorical | Negative |
| | Gender | 0.2589 | | Categorical | |
| | Primary Farmer | 0.1813 | | Categorical | |
| | Conservation Program Interest | <0.0001 ** | $r^2 = 0.1084$ | Categorical | Positive |
| | Farm Income | 0.2443 | | Categorical | |
| | Marginal Land | 0.0153 * | $r^2 = 0.0312$ | Categorical | Positive |
| | Acres | 0.578 | | Continuous | |
| | Beginning Farmer | 0.3193 | | Categorical | |
| Multi Silvopasture | Age | 0.0018 ** | $r^2 = 0.0550$ | Categorical | Negative |
| | Gender | 0.9665 | | Categorical | |
| | Primary Farmer | 0.0651 | | Categorical | |
| | Conservation Program Interest | <0.0001 ** | $r^2 = 0.0718$ | Categorical | Positive |
| | Farm Income | 0.0749 | | Categorical | |
| | Marginal Land | 0.3528 | | Categorical | |
| | Acres | 0.2515 | | Continuous | |
| | Beginning Farmer | 1 | | Categorical | |

**Table 8.** *Cont.*

| Design | Independent Factor [1] | *p*-Value | Model Fit | Variable Type [2] | Relationship [3] |
|---|---|---|---|---|---|
| Forest Farm | Age | <0.0001 ** | $r^2 = 0.0819$ | Categorical | Negative |
| | Gender | 0.0823 | | Categorical | |
| | Primary Farmer | 0.042 * | $r^2 = 0.0159$ | Categorical | Negative |
| | Conservation Program Interest | <0.0001 ** | $r^2 = 0.1620$ | Categorical | Positive |
| | Farm Income | 0.0438 * | $r^2 = 0.0523$ | Categorical | Negative |
| | Marginal Land | 0.0573 | | Categorical | |
| | Acres | 0.0219 * | $r^2 = 0.0193$ | Continuous | Positive |
| | Beginning Farmer | 0.807 | | Categorical | |

Table results from one-way ANOVA in SAS proc General Linear Model (GLM). [1] Independent factors are demographic information of the farmer and his/her farm. Dependent factors include: "would plant the agroforestry design", "would plant the agroforestry design with conservation program funding", and "would plant the design with technical assistance", Age (<35, 35–66, 67+), Gender (Male, Female, Other), Primary Farmer (Yes/No), Conservation Program Interest (Yes/No), Farm Income (<$1000, $1000–$19,999, $20,000–$39,999, $40,000–$69,999, $70,000–$99,000, $100,000+), Marginal Land (Yes/Unsure/No), Acres owned, Beginning Farmer (Yes/No). [2] Classification of independent factor used to predict willingness to plant agroforestry designs, either categorical or continuous. [3] Relationship between independent variable and its influence on dependent variable willingness to plant agroforestry design for significant variables. * Indicates significant *p*-value. ** Indicates highly significant *p*-value. Model fit ($r^2$) is shown for only significant variables.

Landowners' goals for their farm also factor into their management choices [28,49]. We asked landowners to rate the importance of several common agricultural land goals found in the literature, including production for generating income, supporting natural resource conservation, providing recreational opportunities, education and experimental plantings, supporting agritourism, and providing a rural lifestyle [9,20,28,39,50,51]. The top three most important farm goals found in this study were providing a rural lifestyle for self/family, supporting conservation, and production for income (Figure 5). Providing educational experiences or agritourism opportunities were rated as less important, while recreational opportunities were equally important among respondents.

We found that landowners' goals for their farms influence their willingness to plant agroforestry on the land they own. Previous research noted similar relationships between landowner goals and their farm practices [19,49]. Our findings mirror the conclusions of Barbieri and Valdivia [28], who found that landowners with an experience-oriented goal (including conservation, recreation, education, and agritourism) had greater willingness to plant agroforestry (Table 9). Landowners who expressed conservation as an important goal were more willing to plant agroforestry than those who rated it as a low priority. The conservation ethic is an essential factor leading to willingness to adopt that has emerged in several other studies [2,37,47]. In many cases, it is more influential to a landowner or farmer's decision to adopt agroforestry than the financial benefits of conservation program payments or market opportunities [14]. Perhaps most notably, across all planting types, the addition of conservation program funding or technical assistance increased landowners' willingness to plant multifunctional agroforestry designs. This suggests that providing the benefits of either financial help or technical knowledge to landowners would make them more willing to plant agroforestry than establishing the plantings on their own [39].

Interestingly, some of the factors we explored that had no significant influence on willingness to plant agroforestry were found to be important in other adoption studies. While we observed no significant influence of farm size on the landowner's willingness to plant agroforestry, other researchers have found larger farms to be more willing to invest in conservation and plant agroforestry because they have more land and capital available to invest [12,22–24]. Income was another variable that showed no significant relationship with willingness to plant agroforestry. These findings contrast other studies that noted that farm income influenced adoption [22–24]. We may have observed this difference due to our sample being mostly retired farmers who are no longer earning money by farming their land directly. Farming as their primary occupation and farming experience did not have

any significant effect on landowners' willingness to plant agroforestry for our population of landowners. This result is interesting, as other research has noted full-time farmers who relied on their farm as a primary source of income were less interested in agroforestry [52].

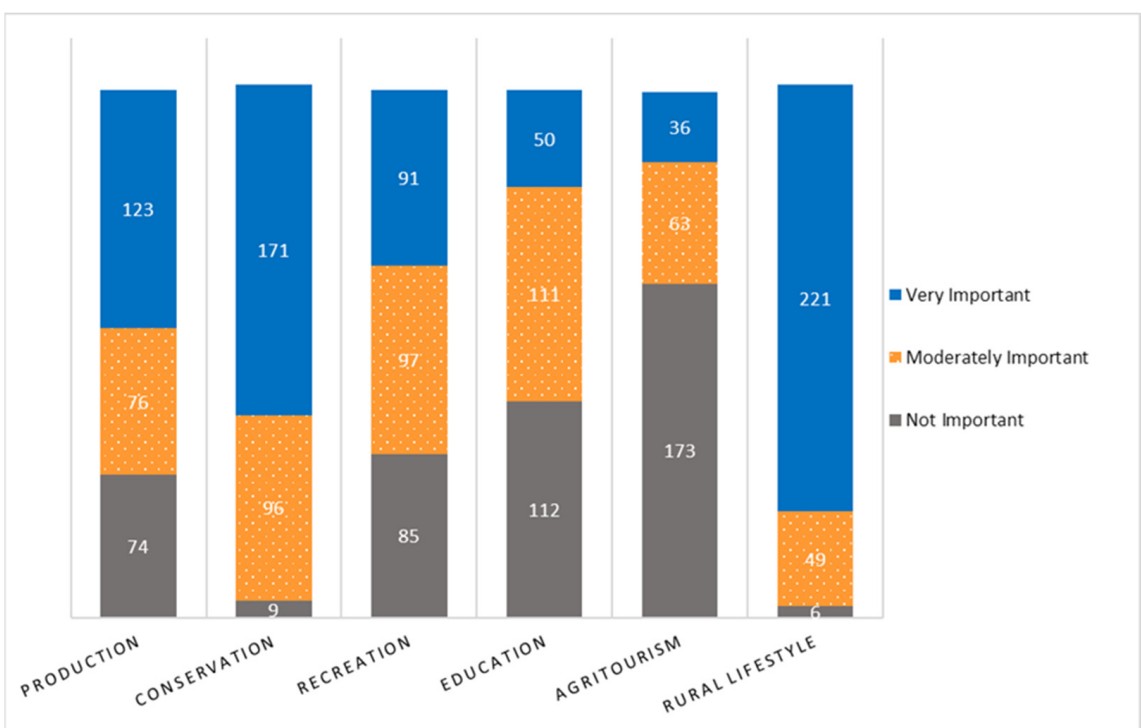

**Figure 5.** Summary of landowners' ranking of important farm goals. Providing a rural lifestyle for themselves or their family was the goal listed as very important by most (221) of the respondents. Providing agritourism opportunities was listed as least important among nearly half of participants (173).

**Table 9.** The influence of landowner farm goals on willingness to plant different multifunctional agroforestry designs, where designs refer to each of the multifunctional agroforestry planting plans shown to participants of the survey.

| Design | Independent Factor [1] | *p*-Value | Model Fit | Variable Type [2] | Relationship [3] |
|---|---|---|---|---|---|
| Multi Windbreak | Goal of Income | 0.1181 | | Categorical | |
| | Goal of Conservation | <0.0001 ** | $r^2 = 0.0954$ | Categorical | Positive |
| | Goal of Recreation | 0.0007 ** | $r^2 = 0.0540$ | Categorical | Positive |
| | Goal of Education | <0.0001 ** | $r^2 = 0.1097$ | Categorical | Positive |
| | Goal of Agritourism | <0.0001 ** | $r^2 = 0.0853$ | Categorical | Positive |
| | Goal of Lifestyle | 0.0528 | | Categorical | |
| Multi Riparian Buffer | Goal of Income | 0.5839 | | Categorical | |
| | Goal of Conservation | 0.0083 ** | $r^2 = 0.0357$ | Categorical | Positive |
| | Goal of Recreation | 0.022 * | $r^2 = 0.0288$ | Categorical | Positive |
| | Goal of Education | <0.0001 ** | $r^2 = 0.0708$ | Categorical | Positive |
| | Goal of Agritourism | 0.0003 ** | $r^2 = 0.0594$ | Categorical | Positive |
| | Goal of Lifestyle | 0.0334 * | $r^2 = 0.0255$ | Categorical | Positive |
| Multi Silvopasture | Goal of Income | 0.12 | | Categorical | |
| | Goal of Conservation | 0.0029 ** | $r^2 = 0.0438$ | Categorical | Positive |
| | Goal of Recreation | 0.0473 * | $r^2 = 0.0234$ | Categorical | Positive |
| | Goal of Education | <0.0001 ** | $r^2 = 0.0895$ | Categorical | Positive |
| | Goal of Agritourism | 0.0019 ** | $r^2 = 0.0476$ | Categorical | Positive |
| | Goal of Lifestyle | 0.0072 ** | $r^2 = 0.0372$ | Categorical | |

**Table 9.** *Cont.*

| Design | Independent Factor [1] | *p*-Value | Model Fit | Variable Type [2] | Relationship [3] |
|---|---|---|---|---|---|
| Forest Farm | Goal of Income | 0.02 * | $r^2 = 0.0294$ | Categorical | Negative |
| | Goal of Conservation | <0.0001 ** | $r^2 = 0.0693$ | Categorical | Positive |
| | Goal of Recreation | 0.0006 ** | $r^2 = 0.0546$ | Categorical | Positive |
| | Goal of Education | 0.005 ** | $r^2 = 0.0397$ | Categorical | Positive |
| | Goal of Agritourism | <0.0001 ** | $r^2 = 0.0706$ | Categorical | Positive |
| | Goal of Lifestyle | 0.0006 ** | $r^2 = 0.0546$ | Categorical | Positive |

Table results from one-way ANOVA in SAS proc General Linear Model (GLM). Source: Landowner Survey, 2021. [1] Independent variable is each of the goals analyzed separately for their influence on the dependent variable, "Would Plant Design", Would Plant with Funding", and "Would Plant with Technical Assistance". [2] Classification of independent factor used to predict willingness to plant agroforestry designs, either categorical or continuous. [3] Relationship between independent variable and its influence on the dependent variable—willingness to plant agroforestry design for significant variables. * Indicates significant *p*-value. ** Indicates highly significant *p*-value. Model fit ($r^2$) is shown for only significant variables.

*3.5. Limitations*

While we can draw meaningful conclusions from this study, there are limitations to this research. We had a low survey response rate of 6% (*n* = 366), which reflects the wider challenge of poor survey response rates from rural populations. The decreasing survey participation is most likely due to the oversampling of this group in academic and census work [33]. As with any survey, it is also vital to consider nonresponse bias [34]. Landowners who are extremely unfavorable to multifunctional plantings may not have taken the time to complete the survey. Other reasons that may have impacted who responded are distrust of the university and government organizations [34,46]. We also must consider that we are trying to quantify preferences, determine future behavior based on current reported opinions, and make broad statements about a population. These findings are only a small portion of the landowner views in the counties we sampled from, and care must be taken when generalizing to the larger landowner population.

*3.6. Future Research*

Moving forward, exploring how to best connect with young landowners who are more open to implementing agroforestry practices on their farm will be important to increase adoption of these practices. Building these connections will include educational programming targeted to younger audiences [53]. Additionally, establishing preferred information sources will be essential to developing the education and outreach programs needed to support productive conservation [54,55]. Beyond individual behavior changes, we need to work to shift government policy to be more supportive of conservation and provide the long-term funding needed to establish and maintain perennial conservation practices [5]. In addition to policy, expanding market research and development will be required to build confidence for farmers to invest in tree and shrub crops [22,24].

**4. Conclusions**

Our survey results suggest that landowners are receptive to agroforestry plantings and rate them higher, on average, than traditional agricultural land management practices. The inclusion of technical assistance or federal conservation program funding was found to increase landowners' willingness to plant multifunctional agroforestry designs. Taken together, this is a promising sign that supporting agroforestry through federal conservation programs will encourage landowners to use these programs, leading to the long-term conservation of important natural resources including soil. Agroforestry practices can provide landowners who are interested in conservation manage their land in ways that support their conservation goals as well as their recreation, education, and lifestyle interests. These findings are helpful for guiding outreach efforts for conservation work and agroforestry adoption. Strengthening conservation-focused extension and educational programing is essential because the knowledge of conservation programs and agroforestry is still a barrier

for landowners to adopt these practices. It will be necessary to replicate similar surveys in other states to gather localized information on landowner goals, interest in conservation programs, and perceptions of agroforestry planting designs relevant to the local farming communities to tailor outreach support to landowners' specific needs.

**Supplementary Materials:** The following supporting information can be downloaded at: https://www.mdpi.com/article/10.3390/land12101911/s1.

**Author Contributions:** Conceptualization, S.L. and R.K.; methodology, R.K., S.L. and B.T.; software, C.S. and R.K.; validation, S.L., M.G. and C.V.; formal analysis, C.S., B.T. and R.K.; investigation, R.K.; resources, S.L.; data curation, R.K.; writing—original draft preparation, R.K.; writing—review and editing, S.L., B.T., M.G., C.V. and S.B.; visualization, R.K.; supervision, S.L. and M.G.; funding acquisition, S.L. All authors have read and agreed to the published version of the manuscript.

**Funding:** This work is supported by the University of Missouri Center for Agroforestry, the USDA/ARS Dale Bumpers Small Farm Research Center agreement number 58-6020-0-007 from the USDA Agricultural Research Service, and the USDA National Institute of Food and Agriculture award number 2018-67019-27853.

**Institutional Review Board Statement:** The research protocol was approved by the University of Missouri Campus Institutional Review Board (IRB Review #300918) and all participants provided informed consent.

**Informed Consent Statement:** Informed consent was obtained from all subjects involved in the study.

**Data Availability Statement:** The data presented in this study are available on request from the corresponding author.

**Acknowledgments:** The authors express their gratitude to Isaac Palomo for his expertise in developing the photo renditions of the agroforestry plantings and to the research participants for giving their time to respond to the survey and share their perspectives of agroforestry.

**Conflicts of Interest:** The authors declare no conflict of interest.

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
