# Peer review of "Survey of Missouri Landowners to Explore the Potential of Woody Perennials to Integrate Conservation and Production"

_land, doi:10.3390/land12101911_

Round 1

Reviewer 1 Report

Manuscript ID: land-2600262-peer-review-v1

Title:  Survey of Missouri Landowners to Explore the Potential of Woody Perennials to Integrate Conservation and Production

REVIEWER REPORT

The study's topic is relevant to the journal's scope and objectives and has scientific significance. The approach described in the manuscript is consistent from a scientific perspective. Conclusions are in line with the evidence and justifications made. There are some issues that need to be developed, though, when the item is evaluated.  I've included a list of my specific criticisms of the paper below.

1.      I think that references must be numbered in order of appearance in the text.

2.      The methodology used in the study is not included in the abstract. It is recommended that the authors briefly mention the methods used in the abstract.

3.      The focus of the article and, thus, the literature the article aims to enrich needs to be sufficiently clear. It is seen that the introduction doesn’t spends enough time outlining the problem. Therefore, the introduction should provide a background on why pest management is important for landowners. Also, the transition from federal conservation programs to agroforestry should be a little more flexible. It is not a very common practice to include tables in the introduction.

4.      The study site, survey details, and statistical Analysis were explained clearly.

5.      The results section was well-organized.

6.      Is the discussion section included with the results in this study? If so, heading 3 (Results) should be Results and Discussion. If not, Discussion should be a separate heading.

Reviewer 2 Report

The core content of this article is about agricultural protection policy and farmland management. The article addresses the economic and resource issues facing U.S. agriculture and the conservation policies and economic support programs the government has established to address them. However, some landowners are not interested in participating in conservation programs due to the inflexibility of management requirements, low program payments, and complex administrative procedures. In addition, farmland already participating in conservation programs is often repurposed for production after the contract expires, rather than maintaining perennial vegetation cover. The article also mentions another option for farmland management—agroforestry, which can simultaneously achieve the goals of farmland protection and agricultural product production. However, the widespread adoption of agroforestry systems in agricultural landscapes still faces several obstacles. Researchers have examined the relationship between landowner decision-making and agroforestry adoption factors, but there are still some hurdles to overcome.

There are some logic problems in the article:

1. Lack of a clear introduction and background introduction: Lack of a clear introduction to the research topic and purpose, which may lead readers to be unclear about the background and goals of the entire article.

2. Lack of detailed description of methods and data: The article mentions some data and findings, but does not provide enough information for readers to understand the research methods and data collection process. This may lead readers to question the reliability and validity of the research.

3. Lack of clarity of arguments and conclusions: The article does not seem to have a clear argument and conclusions section, making it impossible to clearly understand the author's main findings and research conclusions.

4. Many tables in the article seem to be pictures, and the author is requested to make changes, which may violate the principle of plagiarism checking of the article. Among them, the time period in Table 2 is only up to 2002. I have some doubts, is it because the government has not introduced new policies after that?

5. Of the combined 6,708 surveys sent out, 366 responses were collected. Can this data represent the will of the public?

6. In table5, many variables did not pass the significance test, can the author give a more detailed explanation?

Overall, the article needs more careful revision.

Article language, I think only slight errors.

Author Response

Please see the attached report.

Round 2

Reviewer 2 Report

The author made some modifications to the article. I think the authors can try to explain this mechanism in more depth in future studies.
